# Oral prodrug of remdesivir parent GS-441524 is efficacious against SARS-CoV-2 in ferrets

Robert M. Cox [1], Josef D. Wolf[1], Carolin M. Lieber[1], Julien Sourimant [1], Michelle J. Lin [2], Darius Babusis [3], Venice DuPont[3], Julie Chan[3], Kim T. Barrett[3], Diane Lye [3], Rao Kalla[3], Kwon Chun[3], Richard L. Mackman[3], Chengjin Ye [4], Tomas Cihlar[3], Luis Martinez-Sobrido [4], Alexander L. Greninger[2], John P. Bilello [3] & Richard K. Plemper [1✉]

Remdesivir is an antiviral approved for COVID-19 treatment, but its wider use is limited by intravenous delivery. An orally bioavailable remdesivir analog may boost therapeutic benefit by facilitating early administration to non-hospitalized patients. This study characterizes the anti-SARS-CoV-2 efficacy of GS-621763, an oral prodrug of remdesivir parent nucleoside GS-441524. Both GS-621763 and GS-441524 inhibit SARS-CoV-2, including variants of concern (VOC) in cell culture and human airway epithelium organoids. Oral GS-621763 is efficiently converted to plasma metabolite GS-441524, and in lungs to the triphosphate metabolite identical to that generated by remdesivir, demonstrating a consistent mechanism of activity. Twice-daily oral administration of 10 mg/kg GS-621763 reduces SARS-CoV-2 burden to near-undetectable levels in ferrets. When dosed therapeutically against VOC P.1 gamma γ, oral GS-621763 blocks virus replication and prevents transmission to untreated contact animals. These results demonstrate therapeutic efficacy of a much-needed orally bioavailable analog of remdesivir in a relevant animal model of SARS-CoV-2 infection.

[1] Center for Translational Antiviral Research, Institute for Biomedical Sciences, Georgia State University, Atlanta, GA, USA. [2] Virology Division, Department of Laboratory Medicine and Pathology, University of Washington, Seattle, WA, USA. [3] Gilead Sciences Inc, Foster City, CA, USA. [4] Texas Biomedical Research Institute, San Antonio, TX, USA. ✉email: rplemper@gsu.edu

The global COVID-19 pandemic has resulted in far more than 150 million cases and an estimated 3.5 million deaths. The socioeconomic and geopolitical impact of the pandemic is unprecedented and will most likely become fully evident only in postpandemic years. To date, remdesivir is the only small molecule antiviral therapeutic approved by the FDA for the treatment of COVID-19[1]. Originally discovered for respiratory syncytial virus[2], remdesivir is a mono-phosphoramidate prodrug of a ribonucleoside analog that must be administered intravenously[3]. The obligatory parenteral delivery limits access of COVID-19 outpatients to remdesivir treatment, thereby narrowing the therapeutic window in which a direct-acting antiviral targeting an acute viral infection is expected to have its greatest effect[4].

Orally active COVID-19 therapeutics have the potential to maximize patient benefit and ideally prevent progression to severe disease. Several oral therapeutic candidates such as EIDD-2801/molnupiravir[5–7] and AT-527[8] are in advanced stages of clinical trials, but currently, only remdesivir has shown confirmed efficacy for COVID-19 treatment in humans[9]. Following intravenous administration and despite its short plasma half-life, remdesivir is sufficiently stable in non-rodent species to distribute to tissues, such as the lung, where it is rapidly converted to its monophosphate metabolite and then efficiently anabolized to the bioactive triphosphate GS-443902 (Fig. 1a)[2,10,11]. An alternative strategy to generate GS-443902 could be to directly administer the remdesivir parent nucleoside GS-441524, but due to less efficient metabolism of GS-441524 to its monophosphate, higher daily systemic exposures of GS-441524 than those obtained from intravenous remdesivir are required to generate equivalent concentrations of GS-443902 in lung tissue. In addition, GS-441524 has displayed poor oral bioavailability in several species[2,12,13]. In order to obtain high exposures after oral delivery, a prodrug strategy to improve oral absorption was employed leading to the identification of GS-621763, which demonstrated high oral bioavailability in two relevant animal species including non-human primates[2].

SARS-CoV-2 efficiently infects mustelids such as ferrets and mink, and both the direct and reverse zoonotic transmission between mink and humans have been reported[14,15]. Infected ferrets show only mild clinical signs, but the virus readily replicates in the upper respiratory tract and shed virus load in nasal lavages is high, supporting efficient animal-to-animal transmission[6]. Consequently, ferrets recapitulate the presentation of SARS-CoV-2 in the majority of human cases, especially in children and younger adults[16,17].

In this study, we used the ferret model to test the oral anti-SARS-CoV-2 efficacy of GS-621763. The GS-621763 prodrug is presystemically hydrolyzed to afford high systemic exposures of GS-441524 (Fig. 1a). Having determined the plasma pharmacokinetic (PK) profile following oral administration of GS-621763 in ferrets, we examined the effect of oral GS-621763 administered therapeutically against the original SARS-CoV-2 USA-WA1/2020 (WA1/2020) strain and the recently emerged, highly prevalent VOC P.1 (γ) lineage[18,19].

## Results

Prior to in vivo testing, we assessed antiviral potency of both GS-621763 and its metabolite GS-441524 against lineage A isolate WA1/2020 and three recently emerged VOC, hCoV-19/USA/CA_UCSD_5574/2020 ((α), lineage B.1.1.7; CA/2020), hCoV-19-South Africa/KRISP-K005325/2020 ((β), lineage B.1.351; SA/2020), and hCoV-19/Japan/TY7-503/2021 ((γ), lineage P.1; BZ/2021) in cultured cells.

**Antiviral activity against VoCs.** Half-maximal effective concentrations ($EC_{50}$) in SARS-CoV-2-infected Vero E6 cells were highly consistent, ranging from 0.11 to 0.73 μM for GS-621763 (Fig. 1b, Supplementary Table 1) and from 0.11 to 0.68 μM for GS-441524 (Fig. 1c, Supplementary Table 1). Analogous potency ranges were obtained when luciferase-expressing WA1/2020 reporter viruses were examined in dose-response assays in A549 cells stably expressing human ACE2 (A549-hACE2) (Supplementary Table 1)[20]. Toxicity-testing of GS-621763, remdesivir, and GS-441524 in different cell lines and primary human cells derived from different donors revealed half-maximal cytotoxic concentrations ($CC_{50}$) of 40 to >100 μM (Fig. 1d, Supplementary Table 1), 36 to >100 μM (Supplementary Fig. 1, Supplementary Table 1) and >100 μM (Fig. 1e, Supplementary Table 1), respectively, corresponding to selectivity indices ($SI = CC_{50}/EC_{50}$) of GS-621763 > 137 in VeroE6 and >51 in A549-ACE2 cells. The efficacy of GS-441524 and GS-621763 was in parallel assessed on well-differentiated primary human airway epithelium cultures grown at the air-liquid interface and apically infected with VOC γ (Fig. 1f–g). Basolaterally added GS-441524 or GS-621763 displayed similar potency in this disease-relevant human tissue model, returning $EC_{50}$ values of 2.83 and 3.01 μM, respectively. Parallel measurement of transepithelial electrical resistance demonstrated that epithelium integrity was fully preserved at basolateral drug concentrations ≥3 μM.

**Pharmacokinetics following oral administration.** Assessment of GS-621763 plasma PK parameters in the ferret revealed excellent oral bioavailability (Fig. 2a), extensive cleavage presystemically to generate high exposures of GS-441524 in the blood (Supplementary Table 2), efficient distribution to soft tissues of the respiratory system (lung), and confirmed anabolism to bioactive GS-443902 (Supplementary Table 3). Following a single 30 mg/kg oral dose of GS-621763 in ferrets, the daily systemic exposure ($AUC_{0-24h}$) of GS-441524 was 81 μM.h, 4.5 fold higher than the exposure following IV remdesivir at 10 mg/kg and approximately 10-fold greater than that observed following an 200/100 mg IV remdesivir dose in human[21]. Lower levels of bioactive GS-443902 were formed from oral 30 mg/kg GS-621763 dosing compared to 10 mg/kg IV remdesivir (Supplementary Table 3), illustrating the difference in intracellular activation efficiency of the phosphoramidate prodrug remdesivir compared to systemic parent nucleoside GS-441524.

**Prophylactic efficacy in ferrets.** To test antiviral efficacy, we infected ferrets intranasally with $1 \times 10^5$ plaque-forming units (pfu) of WA1/2020, followed by twice daily (b.i.d.) oral treatment with GS-621763 at 20 mg/kg body weight for four days (Fig. 2b). Treatment was initiated at the time of infection, nasal lavages collected in 12-h intervals, and respiratory tissues harvested 4 days after infection. Shed SARS-CoV-2 load in nasal lavages of vehicle-treated animals reached plateau 1.5 days after infection at approximately $1 \times 10^4$ pfu/mL, whereas virus was transiently detectable in lavages of only one ferret of the GS-621763-treatment group at 12 h after infection (Fig. 2c). Clinical signs overall are minor in the ferret model[6]. However, only animals of the vehicle group showed elevated body temperature (Fig. 2d) and reduced weight gain (Fig. 2e). The virus was undetectable in the nasal turbinates extracted from treated animals 4 days after infection, compared to a robust load of approximately $5 \times 10^4$ pfu/g nasal turbinate of animals of the vehicle group (Fig. 2f). Viral RNA copy numbers found in lavages (Fig. 2g) and turbinates (Fig. 2h) mirrored the infectious titer results, revealing a consistent, statistically significant difference between the vehicle and treatment groups of two and three orders of magnitude, respectively. Consistent with prior studies[6], no infectious virions or viral RNA were detectable in the lower respiratory tract (Fig. 2i, j).

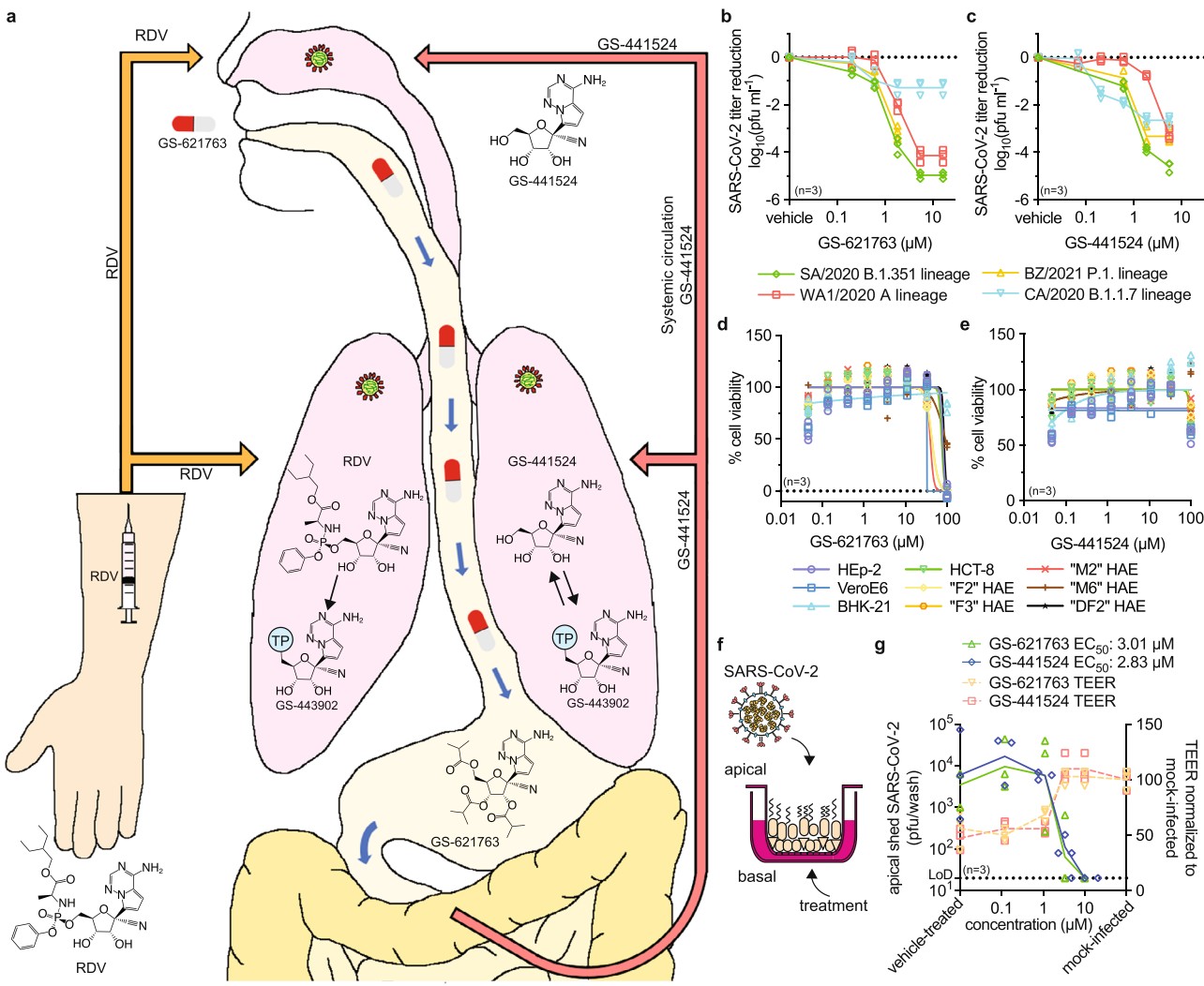

**Fig. 1 Antiviral potency of remdesivir analog GS-621763. a** Schematic depicting metabolism of remdesivir (RDV; orange arrows) and GS-621763 (red/white pills; red arrows) after injection or oral uptake, respectively. Remdesivir distributes into tissues (e.g., lung) and is efficiently metabolized intracellularly to GS-443902 (TP = triphosphate). Conversely, GS-441524 is the dominant plasma metabolite after intestinal absorption of orally administered GS-627163 and is subsequently anabolized to GS-443902 in the tissues. **b–c** Virus yield reduction of SARS-CoV-2 clinical isolates WA1/2020 (red squares), CA/2020 (blue triangles), SA/2020 (green diamonds), and BZ/2021 (yellow triangles) representing the A, B.1.1.7 (α), B.1.351 (β) and P.1 (γ) lineages, respectively, by GS-621763 (b) and GS-441524 (c) on VeroE6 cells. $EC_{50}$ concentrations are specified in Supplementary Table 1. **d–e** In vitro cytotoxicity profiles of GS-621763 (d) and GS-441524 (e) on VeroE6 (blue squares), HEp-2 (purple circles), BHK-21 (light blue triangles), HCT-8 (green triangles) and a panel of primary HAE cells from independent donors ("F2" (yellow diamonds), "F3" (orange circles), "M2" (red "×" symbols), "M6" (brown "+" symbols), "DF2" (black stars)). **f** schematic of well-differentiated air-liquid interface HAE cultures. **g** HAEs were infected from the apical side with SARS-CoV-2 VOC γ and treated from the basolateral side with GS-621763 or GS-441524. Apically shed virus titers (green triangles and blue diamonds for GS-621763 or GS-441524, respectively) on day after infection and the impact of treatment on preserving tissue integrity (transepithelial electrical resistance (TEER)) (orange triangles and red squares for GS-621763 or GS-441524, respectively) are shown. LoD, the limit of detection. In (**b–e, g**), symbols represent individual biological repeats ($n = 3$), lines (**b–c, g**) intersect the mean, lines in (**d–e**) depict nonlinear regression models.

**Therapeutic efficacy and lowest efficacious dose**. To determine the lowest efficacious dose in a clinically more relevant therapeutic setting, we initiated oral treatment 12 h after infection, when the shed virus is first detectable in nasal lavages, at the 10 mg/kg and 3 mg/kg body weight levels, administered b.i.d. (Fig. 3a). EIDD-2801/molnupiravir at 5 mg/kg b.i.d. was given as reference following an identical therapeutic b.i.d. regimen[6]. EIDD-2801 was included as a reference compound, since at the time of study EIDD-2801 was the only nucleoside analog with demonstrated oral efficacy against SARS-CoV-2 in the ferret model. Shed virus load was significantly lower in all treated animals than in the vehicle group within 12 h of treatment onset (Fig. 3b). Virus load in nasal lavages of ferrets receiving GS-

621763 at 3 mg/kg plateaued approximately one order of magnitude lower than in those from vehicle animals, while treatment with GS-621763 at 10 mg/kg or EIDD-2801/molnupiravir reduced shedding to near-detection level by day 3 after infection. Consistent with this inhibitory effect, treatment with 3 mg/kg GS-621763 reduced burden in the turbinates by one order of magnitude (Fig. 3c), while virus burden approached the limit of detection in animals of the 10 mg/kg GS-621763 and EIDD-2801/molnupiravir treatment groups. No significant differences in clinical signs were noted between vehicle animals and any of the treatment groups (Fig. 3d, e).

Viral RNA was detectable in nasal lavages and turbinates of all animals, underscoring efficient infection. However, RNA copies

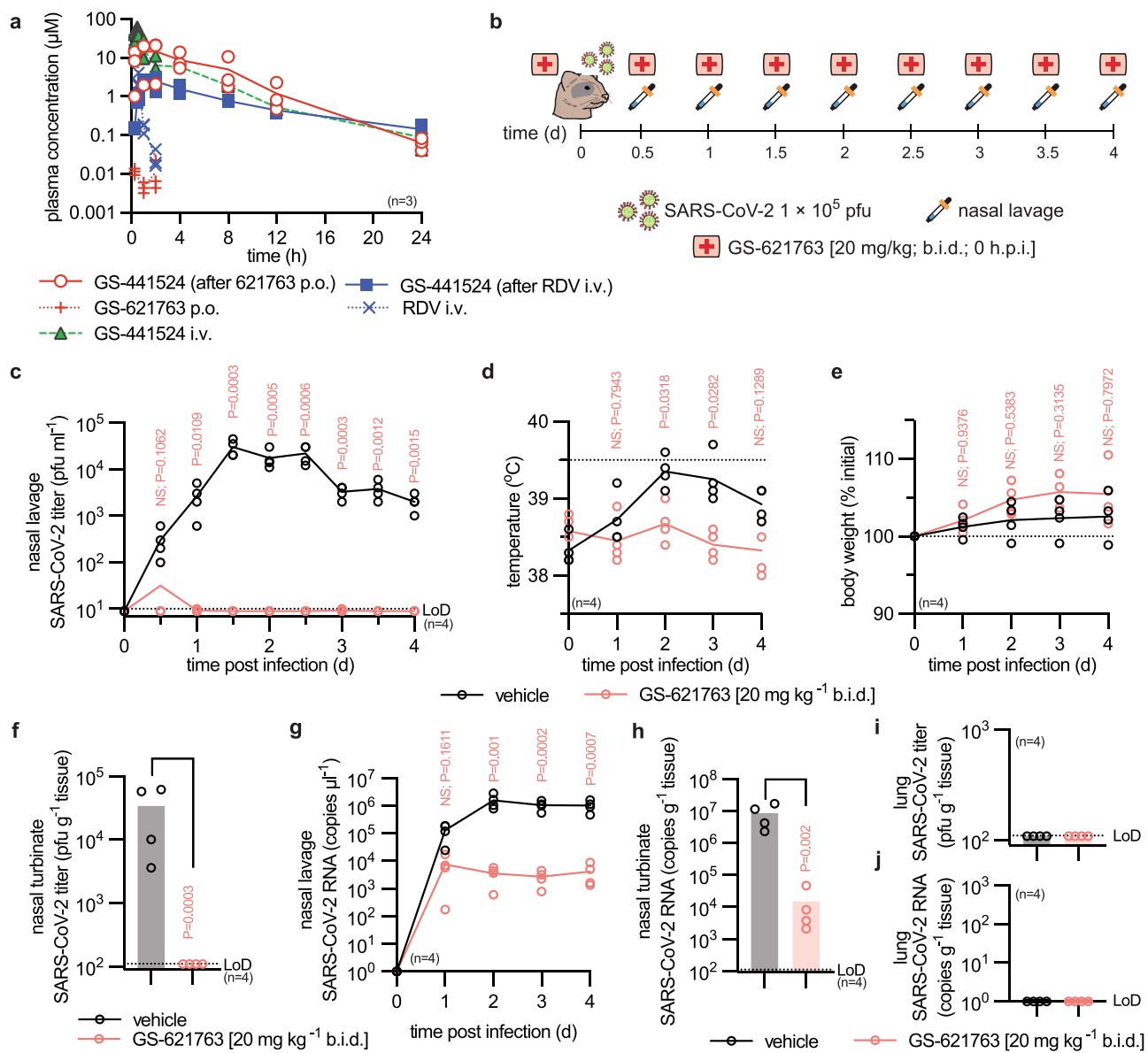

**Fig. 2 Prophylactic efficacy of GS-621763. a** Single-dose PK study in ferrets showing plasma concentrations of GS-441524, GS-621763, and remdesivir (RDV) as specified after dosing with GS-621763 (30 mg/kg; p.o.; red "+" symbols), remdesivir (10 mg/kg; i.v.; blue "×" symbols), and GS-441524 (20 mg/kg; i.v.; green triangles). Symbols represent individual biological repeats ($n = 3$), lines depict sample means. **b**, Schematic of the prophylactic efficacy study design. Ferrets were infected intranasally with $1 \times 10^5$ pfu WA1/2020 (virus symbol). Groups ($n = 4$) were gavaged b.i.d. (first aid symbol) with vehicle or GS-621763 (20 mg/kg) starting at the time of infection. Nasal lavages (pipet symbol) were harvested twice daily. All animals were terminated 4 days after infection. **c** Virus titers from nasal lavages; LoD, limit of detection. **d** Temperature measurements collected once daily. **e** Body weight measured once daily. **f** Infectious titers of SARS-CoV-2 in nasal turbinates harvested four days after infection. **g** SARS-CoV-2 RNA copies present in nasal lavages. **h**, SARS-CoV-2 RNA copies detected in nasal turbinates. **i–j** SARS-CoV-2 infectious particles (i) and SARS-CoV-2 RNA copies (j) in lungs four days after infection. Symbols for vehicle-treated and GS-621763 treated ferrets are shown as black and red circles, respectively (**c–j**). The number of independent biological repeats (individual animals) is shown in each subpanel, symbols represent independent biological repeats, lines (**c–e**, **g**) and bar graphs (**f**, **h–j**) connect or show samples mean, respectively, and P values are stated. 2-way ANOVA with Sidak's post hoc multiple comparison tests (**c–e**, **g**) or two-tailed t test (**f**, **h**).

showed a statistically significant mean reduction in the 10 mg/kg GS-621763 and EIDD-2801/molnupiravir groups compared to vehicle (Fig. 3f, g). These results confirm the oral efficacy of therapeutic GS-621763 against WA1/2020 in a relevant animal model of upper respiratory infection.

**Inhibition of replication and transmission of VOC γ.** To probe the anti-SARS-CoV-2 indication spectrum of GS-621763, we applied the efficacious regimen, 10 mg/kg GS-621763 b.i.d. started 12 h after infection, to recently emerged VOC γ[22] in a combined

efficacy and transmission study (Fig. 4a). After an initial replication delay, the shed virus became detectable in vehicle-treated animals 1.5 days after infection, then rapidly reached a robust plateau of nearly $10^4$ pfu/mL nasal lavage on day 2 after infection (Fig. 4b). Quantitation of viral RNA copies mimicked the profile of the infectious titers, although a low viral RNA load was present in lavages already on the first day after infection (Fig. 4c). Viral titers and RNA copies in nasal turbinates determined 4 days after infection were likewise high, ranging from $10^4$ to $10^5$ pfu/g tissue (Fig. 4d) and $10^8$ to $10^{10}$ RNA copies/g tissue (Fig. 4e), respectively. However, no infectious VOC γ virions or viral RNA were

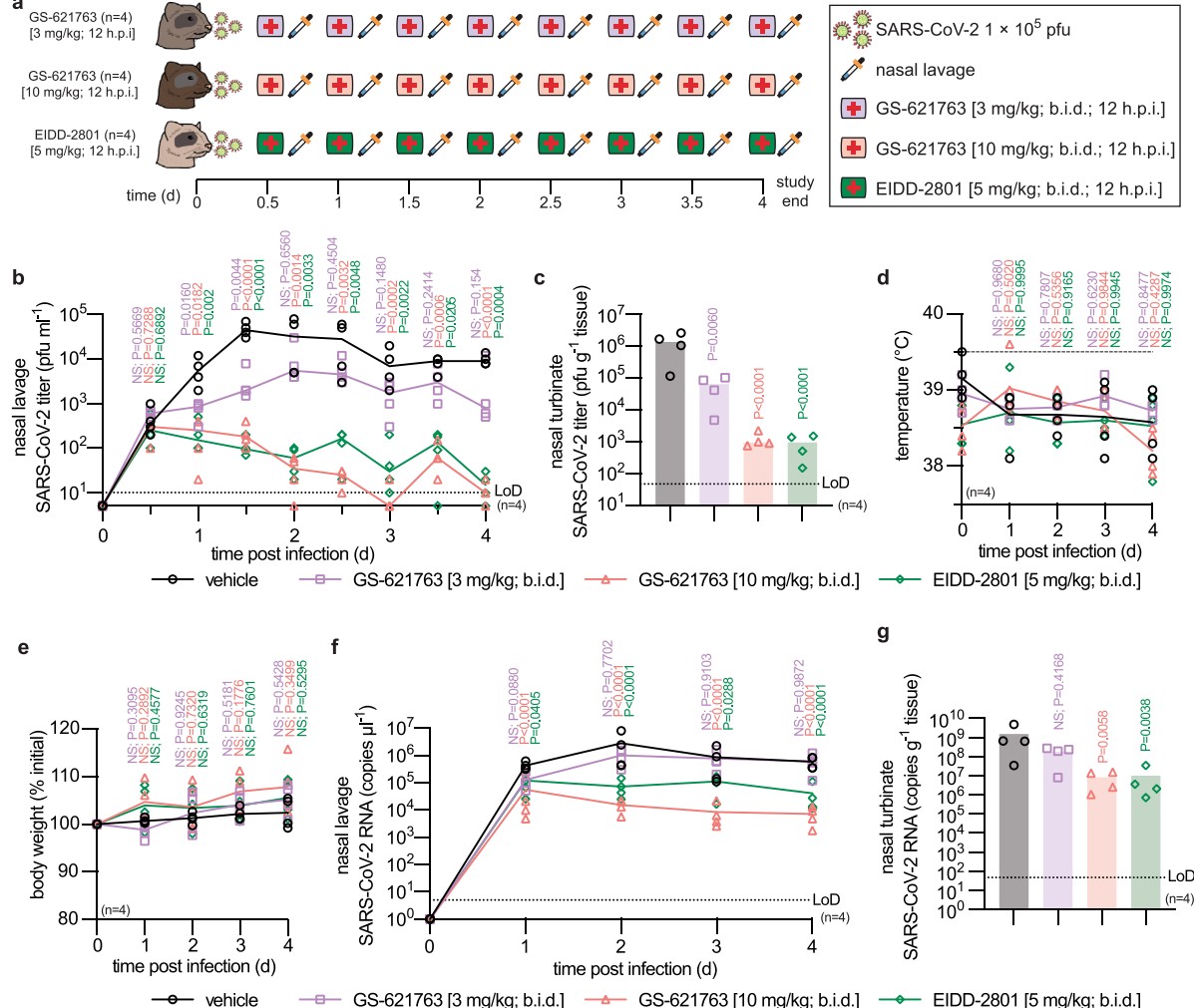

**Fig. 3 Therapeutic efficacy of GS-621763. a** Schematic of the therapeutic efficacy study design. Ferrets were infected intranasally with $1 \times 10^5$ pfu WA1/2020. Symbols as described for Fig. 2a. Starting 12 h after infection, groups of ferrets ($n = 4$) were gavaged b.i.d. with vehicle (black circles in b–g), GS-621763 (3 mg/kg (purple first aid symbol; purple circles in **b–g**) or 10 mg/kg (pink first aid symbol; red circles in **b**)), or treated with EIDD-2801 (5 mg/kg (green first aid symbol; green diamonds in b-g)). Nasal lavages were harvested twice daily. Animals were terminated 4 days after infection. **b** Virus titers from nasal lavages. **c**, Infectious titers of SARS-CoV-2 in nasal turbinates harvested four days after infection. **d** Temperature measurements collected daily. **e** Body weight measured once daily. **f** SARS-CoV-2 RNA copies present in nasal lavages. **g** SARS-CoV-2 RNA copies detected in nasal turbinates. The number of independent biological repeats (individual animals) is shown in each subpanel. Symbols represent independent biological repeats, lines (**b, d, e, f**) and bar graphs (**c, g**) connect or show samples mean, respectively, and $P$ values are stated. 1-way (**c, g**) or 2-way (**b, d, e, f**) ANOVA with Dunnett's post hoc multiple comparison tests. LoD limit of detection.

detected in the lungs of any of these animals (Fig. 4f, g), and no clinical signs such as changes in body weight or fever emerged (Supplementary Fig. 2a, b). This presentation mimicked our previous experience with WA1/2020[6], indicating that VOC γ does not invade the ferret host more aggressively than WA1/2020. Treatment of VOC γ infection with oral GS-621763 was highly efficacious, reducing both shed virus burden and tissue titers to undetectable levels (Fig. 4b, d) and lowering viral RNA copies in nasal lavages and turbinates by over three orders of magnitude (Fig. 4c, e).

Whole genome sequencing of the virus inoculum and virus populations extracted from nasal turbinates confirmed the presence of mutations characteristic for the P.1 VOC[23] (Fig. 4h, Supplementary Dataset 1). In addition, we noted an L260F substitution in nsp6 associated with SARS-CoV-2 adaptation to weasels[14] that had a 60%-allele frequency in the VOC γ inoculum. Four days after infection of ferrets, this mutation had become fully dominant and a second characteristic weasel

mutation, Y453F in the spike protein that was first noted in several clusters of SARS-CoV-2 outbreaks in mink farms[14], had emerged in addition (Fig. 4h, Supplementary Dataset 1). We furthermore noted the presence of an F184V exchange in nsp6 of the VOC γ inoculum, which arose during amplification in VeroE6 cells[23] and was rapidly counterselected against in the ferret host. In contrast, the WA1/2020 inoculum used for our ferret studies did not contain any unreported additional changes (Fig. 4h, Supplementary Dataset 2). WA1/2020 also acquired a weasel-characteristic mutation when passaged through ferrets, N501T in the receptor binding domain of the spike protein[14], but no changes in nsp6 were detected and only one animal harbored a virus population with Y453F substitution in low (8%) allele frequency. Neither the GS-621763-experienced VOC γ nor WA1/2020 populations contained remdesivir resistance mutations previously selected in SARS-CoV-2 (i.e., E802D in nsp12[24]) or the related mouse hepatitis virus[25] (i.e., F476L and V553L in nsp12) nor did any new variants arise in nsp12 at >5% allele

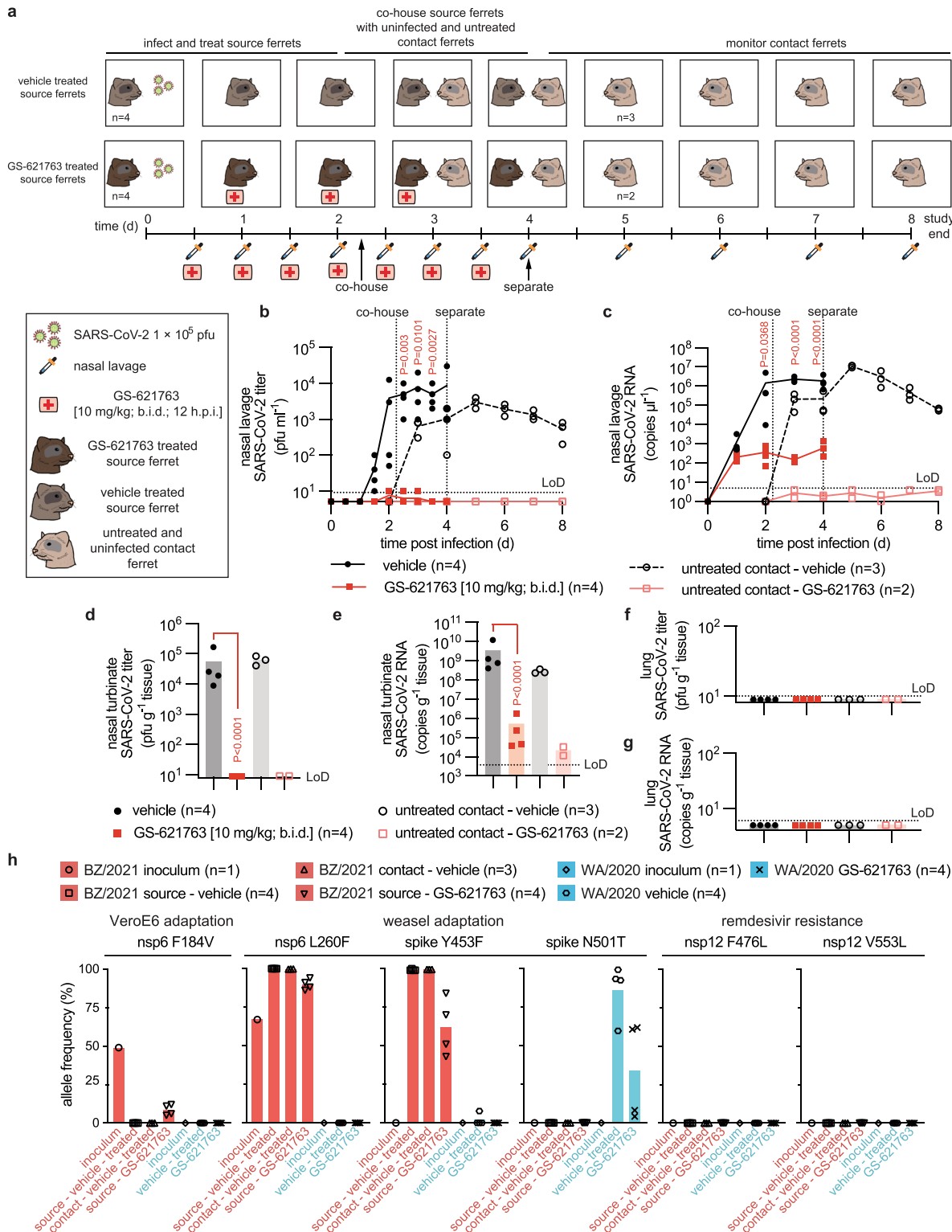

frequency, when viruses were extracted from treated animals at the time of termination (Fig. 4h, Supplementary Datasets 1, 2).

All vehicle-treated animals efficiently transmitted VOC γ to untreated direct-contact ferrets (Fig. 4b–e). Co-housing was started 54 h after infection and continued until termination of the source animals. Shed VOC γ replicated in the contacts without delay, becoming first detectable in nasal lavages within 12 h after initiation of co-housing. This altered replication profile corroborated VOC γ adaptation to the ferret host in the source animals,

and virus populations recovered from contacts of vehicle-treated source animals indeed contained both the L260F exchange in nsp6 and the Y453F mutation in spike (Fig. 4h, Supplementary Dataset 1). Consistent with efficient inhibition of VOC γ replication in the treated source animals by oral GS-621763, treatment completely blocked virus transmission to untreated direct-contact animals. None of the contacts of treated source ferrets shed infectious particles or viral RNA at any time (Fig. 4b, c), infectious viral particles were absent from nasal turbinates 5.5 days after initiation

**Fig. 4 GS-621763 blocks replication and transmission of SARS-CoV-2 VOC γ a, Schematic of the efficacy and contact transmission study design.** All ferrets were infected intranasally with $1 \times 10^5$ pfu BZ/2021 (virus symbol). Symbols as described for Fig. 2a. Groups of ferrets ($n = 4$) were gavaged b.i.d. with vehicle (black circles in **b–g**) or GS-621763 (10 mg/kg; red squares in **b–g**) starting at 12 h after infection. Nasal lavages were harvested twice daily. On the second day after infection, vehicle and GS-621763 ferrets were cohoused with untreated contact ferrets (black open circles and red open squares for vehicle and GS-621763 contact ferrets, respectively). Source ferrets were terminated 4 days after infection and all contact animals were terminated on study day 8. **b** Virus titers from nasal lavages. **c,** SARS-CoV-2 RNA copies present in nasal lavages. **d** Infectious titers of SARS-CoV-2 in nasal turbinates harvested four days after infection. **e** SARS-CoV-2 RNA copies detected in nasal turbinates. **f** Infectious titers of SARS-CoV-2 in lung tissue. **g** SARS-CoV-2 RNA copies present in lung tissue. In (**b–g**), the number of independent biological repeats (individual animals) is shown in each subpanel. Symbols represent independent biological repeats, lines (**b, c**) and bar graphs (**d–g**) connect or show sample means, respectively, and P values are stated. 1-way (**d, e**) or 2-way (**b, c**) ANOVA with Tukey's (**d, e**) or Sidak's (**b, c**) post hoc multiple comparison tests. **h,** Metagenome sequence analysis of inoculum WA1/2020 (blue bar; black diamonds; $n = 1$) and BZ/2021 (red bar; black circles; $n = 1$) viruses, BZ/2021 RNA extracted from ferret nasal turbinates four days after infection (red bars; black squares and downward pointing triangles for vehicle ($n = 4$) and GS-621763 treated ($n = 4$) ferrets, respectively), WA1/202 RNA extracted from ferret nasal turbinates four days after infection (blue bars; black hexagons and "×" symbols for vehicle ($n = 4$) and GS-621763 treated ($n = 4$) ferrets, respectively) and BZ/2021 populations extracted from nasal lavages of contacts of vehicle-treated source animals (red bars; black upward pointing triangles; $n = 3$). Relative allele frequencies of signature residues are shown. Symbols represent independent biological repeats (virus population of individual animals), columns show group means. LoD limit of detection.

---

of co-housing (Fig. 4d), and only a low level of viral RNA ($<10^5$ copies/g nasal turbinate) was detected in nasal turbinates of the contact animals (Fig. 4e).

## Discussion

This study demonstrates oral efficacy of the remdesivir analog GS-621763 against the early clinical SARS-CoV-2 isolate WA1/ 2020 and one of the recently emerged VOC in the ferret model. After oral administration of GS-621763, only low systemic levels of intact prodrug were transiently observed. The parent nucleoside GS-441524 was the major metabolite detected in blood, more than three orders of magnitude higher than intact prodrug, indicating efficient conversion upon intestinal absorption. GS-441524 metabolism inside cells to the same intracellular bioactive triphosphate GS-443902 in tissues links GS-621763 to the well-understood mechanism of action of remdesivir, which blocks SARS-CoV-2 replication by triggering delayed chain termination of the nascent viral RNA chain[26] and/or template-dependent inhibition after incorporation into viral antigenomic RNA[27]. Systemic GS-441524 is less efficient at forming the bioactive GS-443902 in the lungs of ferrets compared to systemic remdesivir, but the exposure of GS-441524 achieved following oral dosing of GS-621763 is sufficient to overcome this difference and generate substantial levels of bioactive GS-443902. The more persistent but lower plasma levels of GS-441524 following intravenous remdesivir are a result of efficient formation of triphosphate in tissues from remdesivir, followed by slow dephosphorylation to GS-441524, which then appears in plasma, as evidenced by its similar half-life with that of the triphosphate GS-443902 in PBMC[21].

Although GS-443902 levels in ferret respiratory tissues after a single 30 mg/kg oral dose of GS-621763 trailed those of intravenous 10 mg/kg remdesivir by approximately fourfold at 24 h, a minimal daily oral dose of 20 mg/kg GS-621763 (10 mg/kg b.i.d.) provided full therapeutic benefit, rapidly reducing shed virus titers to near-undetectable. However, as the GS-443902 active metabolite is quantitated from gross tissue samples, we cannot compare GS-443902 levels resulting from either GS-621763 or remdesivir dosing in distinct respiratory cell types. If human oral bioavailability of GS-621763 is consistent with that observed in ferrets, and the model is predictive of antiviral potential, the dose levels identified in the ferret study as fully efficacious in upper respiratory tract correspond to a feasible human daily dose of approximately 250 mg oral GS-621763.

An observed difference of approximately two to three orders of magnitude between the infectious titers and viral RNA copy numbers in the ferrets closely resembled that described in earlier reports[6,28,29], demonstrating high reproducibility of the model.

At present, the correlation between upper respiratory tract SARS-CoV-2 load at early stages after human infection and the likelihood of progression to viral pneumonia and severe COVID-19 has not been fully understood. However, clinical studies have linked higher upper respiratory virus burden to a heightened patient risk of developing severe COVID-19, requiring intubation or intensive care, and having an unfavorable outcome[30]. Based on these clinical data and the rapid reduction of upper respiratory virus burden seen in the ferret model, we expect that early initiation of oral treatment, ideally immediately following a positive diagnostic test and before the onset of clinical signs[31,32], holds high promise to block viral invasion of the lower respiratory tract and viral pneumonia that necessitates hospitalization.

Recently emerged SARS-CoV-2 variants such as VOC γ are a major concern because of their high prevalence and increased frequency of transmission[18,33]. Ferrets infected with VOC γ did not develop more severe clinical signs than after infection with WA1/2020 and the virus did not advance to the lower respiratory tract, indicating that pathogenicity of this VOC is not fundamentally different from that of WA1/2020 in this model. Peak viral burden and shed virus loads of VOC γ-infected ferrets were equivalent to those of WA1/2020. Reflecting that analysis of circulating SARS-CoV-2 VOC worldwide has yielded no evidence of widespread transmission of remdesivir-resistant strains[24,25,34], GS-621763 and its metabolite GS-441524 were comparably potent against VOC α, β, and γ as against WA1/2020 in cell culture. Importantly, oral GS-621763 was highly efficacious against VOC γ in vivo, reducing infectious titers to near-undetectable levels rapidly after treatment initiation. Whole genome sequencing of GS-621763-experienced viruses recovered from treated ferrets likewise revealed no allele variations in candidate remdesivir resistance sites, indicating that SARS-CoV-2 does not rapidly escape from inhibition by GS-621763 in vivo.

Based on the high transmissibility of VOC γ in the field and very efficient direct-contact transmission of SARS-CoV-2 between ferrets[35], we assessed the impact of oral GS-621763 on VOC γ spread among ferrets. To recapitulate what we consider a realistic scenario for exploring pharmacological interference in community transmission, we therapeutically treated the infected source animals but left their contacts untreated. In this relevant experimental setting, treatment with the same GS-621763 dose of 10 mg/kg b.i.d. completely blocked transmission of VOC γ, resembling the performance of oral EIDD-2801/molnupiravir in the model[6].

These results build confidence that a new generation of orally available broad-spectrum antivirals is emerging that should allow initiation of treatment early after infection and promise to

efficiently interrupt community transmission chains. In addition to immediately contributing to ending the COVID-19 pandemic and containing the continued evolution of increasingly contagious SARS-CoV-2 VOC[36–38], an oral analog of the broad-spectrum antiviral drug remdesivir with confirmed in vivo efficacy may become a cornerstone in the first-line defense against future pandemic threats.

## Methods

**Study design.** Ferrets (*Mustela putorius furo*) were used as an in vivo model to determine efficacy of the orally available remdesivir analog GS-621763 against SARS-CoV-2 infection and transmission to untreated and uninfected contact animals. Oral bioavailability and pharmacokinetic properties of GS-621763 in ferrets were assessed prior to infecting any animals with SARS-CoV-2. For efficacy and transmission studies, anesthetized ferrets were inoculated intranasally with SARS-CoV-2. Nasal lavages were performed periodically at predefined time points to measure virus load. Temperature and body weight were measured once daily for all animals. At four days after infection (study day 8 for contact ferrets), animals were euthanized and nasal turbinates extracted to measure virus load in the upper respiratory tract. SARS-CoV-2 titers were determined by plaque assay and viral RNA copies quantified by RT-qPCR.

**Cells and viruses.** African green monkey kidney VeroE6 (ATCC®, cat# CRL-1586™), human lung adenocarcinoma epithelial Calu-3 (ATCC® HTB-55™), human epithelial/HeLa contaminant HEp-2 (ATCC®, cat# CCL-23™), and baby hamster kidney BHK-21 (ATCC®, cat# CCL-10™) cells were cultivated in a humidified chamber at 37 °C and 5% $CO_2$ in Dulbecco's Modified Eagle's medium (DMEM) (Corning, cat# 10-013-CV, lot# 05721000) supplemented with 7.5% (10% for Calu-3) heat-inactivated fetal bovine serum (FBS) (Corning, cat# 35-010-CV, lot# 14020001). Human epithelial colon adenocarcinoma HCT-8 cells (ATCC® cat# CCL-244™ lot# 70036111) were cultivated at 37 °C and 5% $CO_2$ in Roswell Park Memorial Institute (RPMI-1640) medium (Quality biological, cat# 112-024-101, lot# 723411) supplemented with 2 mM L-glutamine (Gibco, cat# 23030-081) and 10% heat-inactivated FBS.

A549-hACE2 cells that stably express human angiotensin-converting enzyme 2 (hACE2) were grown in the culture medium supplemented with 10 µg/mL Blasticidin S. Primary human airway epithelial (HAE) cells from multiple donors were cultivated at 37 °C and 5% $CO_2$ in Bronchial Epithelial Cell Growth Medium (BEGM) BulletKit following the provider's instructions (Lonza, cat# CC-3171 lot# 0000889952 with supplement cat# CC-4175 lot# 0000848033). Human Bronchial Tracheal Epithelial cells (HBTEC) were derived from the following donors: "F2" from a 29-year old Caucasian female (Lifeline, cat# FC-0035, lot# 5101); "F3" from a 42-year old Caucasian female (Lonza, cat# CC-2540S, lot# 0000519670); "M2" from a 40-year old Caucasian male (Lonza, cat# CC-2540S, lot# 0000667744); and "M6" from a 48-year old Caucasian male (Lonza, cat# CC-2540S, lot# 0000544414). Diseased (Asthma) Human Bronchial Epithelial (DHBE) cells "DF2" were from a 55-year old Caucasian female (Lonza, cat# 00194911 S, lot# 0000534647). Primary HAE was used for cytotoxicity assays at passage ≤3. Cell lines were routinely checked for mycoplasma and bacterial contamination.

SARS-CoV-2 strains were obtained from BEI and propagated using Calu-3 cells supplemented with 2% FBS in accordance with approved biosafety level 3 protocols. Virus stocks were stored at −80 °C. Stock virus titers were determined by plaque assay and stocks authenticated through metagenomic sequencing.

**Plaque assays.** Vero E6 cells were seeded in 12-well plates at $3 \times 10^5$ cells per well. The following day, samples were serially diluted in DMEM containing Antibiotic-Antimycotic (Gibco) supplemented with 2% FBS. Dilutions were then added to cells and incubated for 1 h at 37 °C. Cells were subsequently overlaid with 1.2% Avicel 581-NF (FMC BioPolymer) in DMEM containing Antibiotic-Antimycotic (Gibco) and allowed to incubate for 3 days at 37 °C with 5% $CO_2$. After 3 days, the overlay was removed, cells were washed once with phosphate buffered saline (PBS) and fixed with neutral buffered formalin (10%) for 15 min. Plaques were then visualized using 1% crystal violet followed by washing with water.

**Compound sources and chemical synthesis.** Remdesivir was either purchased from MedChemExpress (cat# HY-104077, batch# 46182) or synthesized at Gilead Sciences, Inc. GS-441524 was either purchased from MedChemExpress (cat# HY-103586, batch# 62110) or synthesized at Gilead Sciences Inc. GS-621763 was synthesized at Gilead Sciences Inc:[2] [1]H NMR (300 MHz, CHCl$_3$-$d_3$) δ 11.15 (bs, 1H), 8.27 (bs, 1H), 7.95 (s, 1H), 7.32 (m, 1H), 7.07 (m, 1H), 6.05 (d, *J* = 6.0 Hz, 1H), 5.44 (t, *J* = 5.1 Hz, 1H), 4.66 (t, *J* = 3.6 Hz, 1H), 4.32 (m, 2H), 2.73–2.52 (m, 3H), 1.27–1.14 (m, 18H). LC-MS (B) *m/z* = 502.2 [M + H], 500.1 [M − H]. All EIDD-2801/molnupiravir used in this study was provided by Gilead Sciences Inc., sourced from MedChemExpress.

**Cytotoxicity assays.** In each well of 96-well plates, 7500 cells were seeded (Corning, cat# 3598). Cells were incubated with threefold serial dilutions of compound from a 100 µM maximum concentration. Each plate included 4 wells of positive (100 µM cycloheximide (Millipore Sigma, cat# C7698-5G)) and negative (vehicle (0.2% dimethyl sulfoxide (DMSO)) controls for normalization. Plates were incubated in a humidified chamber at 37 °C and 5% $CO_2$ for 72 h. PrestoBlue™ Cell Viability Reagent (ThermoFisher Scientific, cat# A13262) was added in each well (10 µl/well) and fluorescence recorded on a Synergy H1 multimode microplate reader (BioTek) after 1-h incubation (excitation 560 nm, emission 590 nm). Raw data was normalized with the formula: % cell viability = 100 × (signal sample—signal positive control)/(signal negative control—signal positive control). Fifty percentage cytotoxic concentrations ($CC_{50}$) and 95% confidence intervals after nonlinear regression were determined using the inhibitor vs normalized response equation in Prism 9.1.0 for MacOS (GraphPad). For cytotoxicity assays in A549-hACE2 cells, compounds (200 nl) were spotted onto 384-well plates prior to seeding 5000 A549-hACE2 cells/well in a volume of 40 µl culture medium. The plates were incubated at 37 °C for 48 h with 5% $CO_2$. On day 2, 40 µl of CellTiter-Glo (Promega) was added and mixed 5 times. Plates were read for luminescence on an Envision (PerkinElmer) and $CC_{50}$ values calculated using a nonlinear four parameter regression model.

**Virus yield reduction.** In 12-well plates 16 h before infection, $2 \times 10^5$ VeroE6 cells were seeded per well. Confluent monolayers were then infected with the indicated virus at a multiplicity of infection (MOI) of 0.1 pfu/cell for 1 h at 37 °C with frequent rocking. Inoculum was removed and replaced with 1 mL of DMEM with 2% FBS and the indicated concentration of compound. Cells were incubated at 37 °C and 5% $CO_2$ for 48 h. Supernatant were harvested, aliquoted and stored at −80 °C before being analyzed by plaque assay.

**Reporter virus assays.** A549-hACE2 cells (12,000 cells per well in medium containing 2% FBS) were plated into a white clear-bottomed 96-well plate (Corning) at a volume of 50 µl. On the next day, compounds were added directly to cultures as 3-fold serial dilutions with a Tecan D300e digital liquid dispenser, with DMSO volumes normalized to that of the highest compound concentration (final DMSO concentration < 0.1%). The diluted compound solutions were mixed with 50 µl of SARS-CoV-2-Nluc (MOI 0.025 pfu/cell), expressing a nano luciferase reporter protein (kind gift of Xuping Xie and Pei-Yong Shi (University of Texas Medical Branch; Galveston, TX)). At 48 h postinfection, 75 µl Nano luciferase substrate solution (Promega) was added to each well. Luciferase signals were measured using an Envision microplate reader (Perkin Elmer). The relative luciferase signals were calculated by normalizing the luciferase signals of the compound-treated groups to that of the DMSO-treated groups (set as 100%). $EC_{50}$ values were calculated using a nonlinear four parameter variable slope regression model.

**Air-liquid interface (ALI) human airway epithelial cells (HAE) shed viral titer reduction.** Approximately 150,000 viable "F3" cells per cm[2] at passage three were seeded on Transwell 6.5 cm polyester membrane insert with 0.4 µm pore size (Corning, cat# 3470). Upon reaching confluence (day 4 postseeding), basal media was removed and replaced with PneumaCult-Ex Plus (Stemcell Technologies cat# 05040), while apical media was removed to create an air-liquid interface. Beating ciliae, mucus production and transepithelial electrical resistance (TEER) > 300 Ohm*cm[2] were noticeable ~3 weeks post ALI, confirming successful differentiation. Cells were maintained in a differentiated state with weekly apical washes of mucus with PBS for 5 months before infection. One hour prior to infection, TEER was measured with 150 µl PBS and basal media was replaced with fresh media containing indicated serial dilutions (threefold down from 10 µM) of GS-621763 or GS-441524 or vehicle (dimethylsulfoxide 0.1%). The apical side was infected with ~25,000 PFU of SARS-CoV-2 gamma isolate grown on Calu-3 cells (lineage P.1., isolate hCoV-19/Japan/TY7-503/2021 (BZ/2021; Brazil P.1), BEI cat# NR-54982) in 100 µl DMEM for 1 h at 37 °C, then the inoculum was removed and washed with PBS twice. Cells were incubated for 3 days at 37 °C before final TEER measure and fixation with 10 % neutral buffered formalin for 1 h. Shed apical viruses were harvested with 200 µl PBS for 30 min at 37 °C 48 and 72 h postinfection and viral titers were estimated by plaque assay. To determine EC50s, log viral titers were normalized using the average top plateau of viral titers to define 100% and were analyzed with a nonlinear regression with the variable slope with Prism 9.0.1 for MacOS (GraphPad). TEER were measured with the EVOM or EVOM3 system (World Precision Instruments).

**Pharmacokinetics.** Female ferrets were either intravenously administered 10 mg/kg remdesivir as a 30-min infusion or orally administered 30 mg/kg GS-621763, after which plasma was isolated at 7–9 timepoints postadministration. Plasma samples underwent methanol protein precipitation followed by centrifugation. The resulting supernatants were isolated, evaporated to dryness under nitrogen and reconstituted with 5% acetonitrile for injection onto an LC-MS/MS system (Sciex API-4500). Concentrations of remdesivir, GS-621763, and GS-441524 were determined using 9-point calibration curves spanning at least 3 orders of magnitude, with quality control samples to ensure accuracy and precision, prepared in normal ferret plasma. Analytes were separated by a 50 × 3.0 mm, 2.55 µm Synergi Polar-RP 30 A column (Phenomenex, Inc.) using a mobile phase A consisting of 10 mM ammonium formate with 0.1% formic acid and a mobile phase B consisting

of 0.1% formic acid in acetonitrile. A multi-stage linear gradient from 5% to 95% mobile phase B at a flow rate of 1 mL/min was employed for analyte separation (Shimadzu). Pharmacokinetic parameters were calculated using Phoenix Win-Nonlin (version 8.2, Certara) and concentration-time profiles generated using Prism (version 8, GraphPad). Ferret lungs were collected at 24 h following initiation of drug administration. Whole tissues were quickly isolated and immediately placed into liquid nitrogen and stored at -80 °C until processing and LC-MS/MS analysis[2]. Reported values for lung total nucleosides are the sum of (GS-441524 and mono-, di-, and triphosphate (GS-443902) metabolites).

**Ferret efficacy studies.** Female ferrets (6–10 months old, *Mustela putorius furo*) were purchased from Triple F Farms. Ferrets were rested for 7 days after arrival. Ferrets were then housed individually or in groups of 2 in ventilated negative-pressure cages in an ABSL-3 facility. Based on the previous experiments[6], ferrets were randomly assigned to groups ($n = 4$) and used as an in vivo model to examine the efficacy of orally administered compounds against SARS-CoV-2 infection. No blinding of investigators was performed. Ferrets were anesthetized using dexmedetomidine/ketamine and infected intranasally with $1 \times 10^5$ pfu 2019-nCoV/USA-WA1/2020 in 1 mL (0.5 mL per nare). Body weight and rectal temperature were measured once daily. Nasal lavages were performed twice daily using 1 mL sterile PBS (containing Antibiotic-Antimycotic (Gibco)). Nasal lavage samples were stored at -80 °C until virus titration could be performed by plaque assay. Treatment (once daily (q.d.) or twice daily (b.i.d.)) was initiated at either 0 or 12 h after infection and continued until 4 days postinfection with either vehicle (2.5% dimethyl sulfoxide; 10% Kolliphor HS-15; 10% Labrasol; 2.5% propylene glycol; 75% water) or compound. Four days after infection, ferrets were euthanized, and tissues and organs were harvested and stored at -80 °C until processed.

**Contact transmission in ferrets.** Eight ferrets were anesthetized and inoculated intranasally with $1 \times 10^5$ pfu of hCoV-19/Japan/TY7-503/2021. Twelve hours after infection, ferrets were split into two groups ($n = 4$; 2 ferrets per cage) and treated with vehicle or GS-621763 (10 mg kg$^{-1}$) twice daily (b.i.d.) via oral gavage. At 54 h after infection, uninfected and untreated contact ferrets (two contacts for GS-621763; three contacts for vehicle) were co-housed with source ferrets. Co-housing was continued until 96 h after infection and source ferrets were euthanized. Contact ferrets were housed individually and monitored for an additional 4 days after separation from source ferrets and subsequently euthanized. Nasal lavages were performed on all source ferrets every 12 h and all contact ferrets every 24 h. For all ferrets, nasal turbinates and lung tissues were harvested to determine viral titers and the detection of viral RNA.

**SARS-CoV-2 titration in tissue extracts.** Selected tissues were weighed and mechanically homogenized in sterile PBS. Homogenates were clarified by centrifugation ($2000 \times g$) for 5 min at 4 °C. Clarified homogenates were then serially diluted and used in plaque assays to determine virus titer as described above.

**Quantitation of SARS-CoV-2 RNA copy numbers.** To probe viral RNA in selected tissues, samples were harvested and stored in RNAlater at $-80$ °C. Total RNA from tissues was isolated using a RNeasy mini kit (Qiagen), in accordance with the manufacturer's protocol. For nasal lavage samples, total RNA was extracted using a ZR viral RNA kit (Zymo Research) in accordance with the manufacturer's protocol. SARS-CoV-2 RNA was detected using the nCoV_IP2 primer-probe set (National Reference Center for Respiratory Viruses, Pasteur Institute) (Supplementary Table 4). An Applied Biosystems 7500 using the StepOnePlus real-time PCR system was used to perform RT-qPCR reactions. The nCoV_IP2 primer-probe set was using in combination with TaqMan fast virus 1-step master mix (Thermo Fisher Scientific) to detect viral RNA. SARS-CoV-2 RNA copy numbers were calculated using a standard curve created from serial dilutions of a PCR fragment (12669-14146 nt of the SARS-CoV-2 genome). For RNA copies in tissue samples, RNA copies were normalized to the weights of the tissues used.

**Next-generation sequencing.** To authenticate virus stocks, metagenomic sequencing was performed as described[39,40], while to sequence lower viral load in vivo samples the COVID-Seq (Illumina) amplicon tiling protocol was used. For metagenomic sequencing, viral RNA was treated with Turbo DNase I (Thermo Fisher), converted to cDNA using random hexamers and SuperScript IV reverse transcriptase, and double-stranded cDNA created using Sequenase v2.0. Sequencing libraries for both sets of libraries were generated using Nextera Flex (Illumina) and cleaned using 0.8 × Ampure XP beads and pooled equimolarly before sequencing on an Illumina 1 × 100 bp NextSeq2000 run. Raw fastq reads were adapter- and quality-trimmed with Trimmomatic v0.39[41]. To interrogate potential resistance alleles, reference-based mapping to NC_045512.2 was carried out using our modified Longitudinal Analysis of Viral Alleles (LAVA—https://github.com/michellejlin/lava)[42] pipeline. LAVA constructs a candidate reference genome from early passage virus using bwa[43], removes PCR duplicates with Picard, calls variants with VarScan[44,45], and converts these changes into amino acid changes with Annovar[46]. Consensus sequences were called with TAYLOR[39] and deposited in

NCBI GenBank. Accession numbers (Supplementary Table 5) are as follows: input strain WA1/2020, MZ433205; WA1/2020 recovered from ferrets, MZ433206 - MZ433213; input strain BZ/2021, MZ433225; BZ/2021 recovered from source ferrets, MZ433214 - MZ433221; BZ/2021 recovered from contacts of vehicle-treated source ferrets, MZ433222–MZ433224. Raw reads for these sequences are publicly available on SRA (BioProject PRJNA740065).

**Ethics statement.** All in vivo efficacy studies were conducted at Georgia State University in compliance with the Animal Welfare Act Code of Federal Regulations and the Guide for the Care and Use of Laboratory Animals of the National Institutes of Health. All studies involving SARS-CoV-2-infected ferrets were approved by the Georgia State Institutional Animal Care and Use Committee under protocol A20031. Experiments at Georgia State University using infectious SARS-CoV-2 were performed in BSL-3/ABSL-3 facilities at Georgia State University and approved by the Georgia State Institutional Biosafety Committee under protocol B20016. Experiments at Gilead Sciences, Inc. using infectious SARS-CoV-2 were performed in a BSL-3 facility and approved by an institutional biosafety committee.

**Statistics and reproducibility.** The Microsoft Excel (versions 16.42, 16.43, and 16.48), GraphPad Prism (versions, 8.0, 9.0.1, and 9.1.0), and Numbers (version 10.1) software packages were used for data collection and analysis. One-way or two-way ANOVA with Dunnett's or Tukey's multiple comparisons post hoc test were used to evaluate statistical significance when comparing more than two groups or two independent variables. When comparing two variables, a two-tailed unpaired t-test was performed to determine statistical significance. The specific statistical test used to individual studies is specified in the figure legends. RT-qPCR data were collected and analyzed using the StepOnePlus (version 2.1; Applied Biosystems) software package. Final figures were assembled in Adobe Illustrator (version CS6). All numerical raw data and summaries of individual statistical analyses are provided in supplementary datasets 3 and 4. Effect sizes between groups in the ANOVAs were calculated as $\eta^2 = (SS_{effect})/(SS_{total})$ for one-way ANOVA and $\omega^2 = (SS_{effect} - (df_{effect})(MS_{error}))/MS_{error} + SS_{total}$ for two-way ANOVA ($SS_{effect}$, sum of squares for the effect; $SS_{total}$, the sum of squares for total; $df_{effect}$, degrees of freedom for the effect; $MS_{error}$, mean squared error). The statistical significance level $\alpha$ was set to <0.05 for all experiments. Exact P values are shown in the individual graphs. Appropriate sample sizes were determined using power analyses (GPower 3.1; University of Duesseldorf).

**Reporting Summary.** Further information on research design is available in the Nature Research Reporting Summary linked to this article.

## Data availability

All next-generation sequencing data is publicly available on SRA (BioProject PRJNA740065). All accession codes for sequence data are available in Supplementary Table 5. Sequencing data is summarized in Supplementary Data 1–2. All data generated and analyzed during this study are included in this published article (and its supplementary information files). Source data and statistical analyses for Figs. 1–4, Supplementary Tables 13 and Supplementary Figs. 1–2 are provided with the paper in Supplementary Data 3–4.

## Code availability

This study does not use proprietary codes. All computer codes and algorithms used are specified in the Methods section.

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

## Acknowledgements

We thank BEI Resources for SARS-CoV-2 isolates, X. Xie and P.-Y. Shi for providing SARS-CoV-2-Nluc, A. Shornikov of the Gilead Sample Bank group for compound distribution, the High Containment Core and the Department for Animal Research of Georgia State University for support, and A. L. Hammond and D. Porter for critical reading of the manuscript.

## Author contributions

J.P.B. and R.K.P. coordinated the study. R.M.C. and J.S. performed virus-stock preparations and virus titrations. R.M.C., J.D.W., C.M.L. and R.K.P. performed animal inoculations, nasal lavage sampling, animal necropsies, and/or titration of virus from ferrets R.M.C. extracted RNA from all animal samples and performed all RT-qPCR experiments and analyses. J.S. performed virus reduction assays and cytotoxicity assessments. D.B. coordinated pharmacokinetics studies in ferrets. K.T.B. and D.L. prepared compound formulations. V.D.P., J.P.B. and J.C. performed reporter virus and cytotoxicity assays in ACE2-A549 cells. R.K, K.C. and R.L.M. synthesized and sourced compounds. M.J.L. and A.L.G. performed all next-generation sequencing and analyses. R.M.C. created all figure schematics. R.M.C., J.S., J.P.B. and R.K.P. were responsible for experimental design, data analysis, and data presentation. R.M.C., J.D.W., C.M.L., J.S., D.B., L.S.M., A.L.G., R.L.M., C.Y., J.P.B. and T.C. edited the manuscript. J.P.B. and R.K.P. conceived and designed the study. R.K.P. wrote the manuscript. This work was supported by Gilead Sciences Inc. and, in part, by Public Health Service grants AI153400 (to RKP) and AI141222 (to RKP) from the NIH/NIAID. NIH/NIAID had no role in study design, data collection, and interpretation, or the decision to submit the work for publication.

## Competing interests

All authors affiliated with Gilead Sciences Inc. may hold stock or stock options in Gilead Sciences Inc. R.K.P. was the principal investigator of a Gilead-sponsored research agreement with Georgia State University. He has received funding from Gilead Sciences Inc. to support parts of this work. All other authors declare no competing interests.
