## [Peer Review File · Nature Communications]

Oral prodrug of remdesivir parent GS-441524 is efficacious against SARS-CoV-2 in ferretsReviewers' Comments:

Reviewer #1:

Remarks to the Author:

In this manuscript the authors profile GS-621763, an oral prodrug of the remdesivir parent previously identified within a discovery program targeting RSV. Here the authors report on GS-621763's PK, its efficacy against SARS-CoV-2 isolates in vitro and its efficacy in prophylactic and therapeutic modes in an in vivo ferret SARS-CoV-2 model of infection. Additionally, the authors report that oral dosing of GS-621763 in SARS-CoV-2-infected ferrets prevents transmission of the virus to untreated contact animals for a variant of concern of the P.1 lineage. They also monitor the mutations in SARS-CoV-2 isolates that emerge during their experiments.

This report is of significant interest, as development of effective oral therapies for COVID-19 would have a positive impact on controlling the current pandemic by improving accessibility to antivirals and potentially limit community spread of the infection.

The manuscript is comprehensive; however, several edits, most minor, should be attended to prior to publication:

1. Additional methods or description of data generation for Extended Data Table 3 is missing. What timepoint were the lungs harvested at (24 h post dose?) and what was measured to determine the "lung total nuc" metric?
2. No reference or methods for generation of A549-hACE2 cell line is present in the manuscript. These are also not listed in the Reporting Summary. Please add references/methods as appropriate.
3. Is there an explanation as to why GS-621763 is less efficacious in vitro against the CA/2020 B.1.1.7 lineage, causing only ~1 log reduction vs ~5 log reduction in titer of other strains?
4. Legends for a couple plots in the manuscript refer to symbols that represent individual biological replicates, but instead show a single (mean or a median) point: Fig. 2a, Extended Data Fig. 1.
5. In the introduction, line 59, the authors refer to high oral bioavailability of GS-621763 "in several species". Because "several" technically refers to more than two, and bioavailability seemed to be determined for compound 13 (GS-621763?) in the reference for only rat and cynomolgus monkeys please either list the species or change the wording.
6. Line 33 in abstract: adding that the studies were performed in ferrets would improve readability: e.g., ... to near-undetectable levels in ferrets.
7. Lines 79 – 81, it is unclear from the results section that EC50s in Vero E6 cells are being discussed. Listing Vero E6 in this section of the text would be helpful.
8. In legend for Figure 3, a comma is missing between vehicle and GS-621763 (line 382).
9. Line 494: "were added" is redundant and an extra closing bracket is missing.

Reviewer #2:

Remarks to the Author:

Hereby I would like to congratulate the authors of the manuscript "Oral Prodrug of remdesivir parent GS-441524 is efficacious against SARS-CoV-2 and a variant of concern in ferrets" with their nice paper. I am happy that people are looking into the great potential of GS-441524 and was thus content to review the paper! The authors did a great job studying the antiviral effect of their newly developed prodrug (GS-621763) in cell culture and in ferret. The results look promising; however, I lack some information about crucial points.

Major concerns

- o I'm not entirely convinced that their approach is the most useful. Please comment on the fact that you work with a pro-drug of GS-441524, which still needs to be converted to the active triphosphate,

which is a rate-limiting step in vivo. Your compound is basically a new prodrug of GS-441524 (tri-isopropylester prodrug instead of phosphoramidate prodrug that is Remdesivir), with the advantage of a better oral absorption and thus availability.

I do agree on the advantage compared to Remdesivir (Oral and high systemic levels of GS-441524), but please highlight why it is better to use this pro-drug, compared to GS-441524 directly, as this compound is also known to be orally available and effective in vivo against SARS-CoV-2. Please comment on other possible chemical options, e.g. a prodrug which easily converts to GS-443902 or an oral Remdesivir analogue (although known to be very difficult)?

Similarly, please compare the effect of your pro-drug in vivo with the effect of oral administration of GS-441524 directly. This will highlight to the advantage of your prodrug (higher exposure of GS-441524) compared to GS-441524 itself.

- o PK should include measurement of GS-443902 (in PBMC) for all compounds/routes.

- o I appreciate the study with delayed start of treatment, however, I believe 12hours is not enough. Please comment on further delay (24 – 48h) of treatment start, as this would be more relevant in a clinical setting.

- o Besides all the in vivo work, it maybe not highly relevant, but it would be nice to see antiviral data in multiple cell types instead of only VeroE6 cells. I encourage the group to verify antiviral effect especially in primary HAE cells to have more human data.

Minors concerns

- o Improve the picture (Fig. 1) to describe Remdesivir metabolization in human (monophosphate metabolite and anabolization to GS-443902).

- o Remove all P values from the picture to make it more ordered/easier to view.

- o In the transmission experiment: after 52 hours, there is already a great reduction of infectious virus in the ferret (as shown in the previous experiments). So, maybe it would be better to co-house the animals in an earlier stage (when there is still a lot of virus present).

Although I still believe in the effect of the compound, I think the transmission experiment *persé* is not as relevant as stated. (patients that are being treated are aware that they are infected and thus quarantine) More relevant would be later start of treatment (more than the 12hours in the paper here), as mentioned before.

- o Misspelling of the pro-drug in the legend of Fig. 1 (GS-627163 instead of GS-621763)?

- o The authors claim there is little escape variants after in vivo treatment, however, a study is only 4 days. I feel this should be backed by long term cell culture experiments or longer in vivo studies.

- o Please specify how temperature is measured. I'm not sure if this is a relevant factor, as in Fig. 3,D it shows no temperature elevation in vehicle treated animals.

- o If this prodrug is suitable for clinical studies, please comment on production expectation, stability, storage and if you expect any toxicity issues.

- o Fig. 2,H shows 10^7 RNA copies in nasal turbinate in vehicle treated, while Fig. 3,G shows 10^9 copies. Is this expected variation?

Reviewer #3:

Remarks to the Author:

This study examined the efficacy of the oral prodrug (GS-621763) of remdesivir parent (GS-441524) to reduce COVID-19 viral titers and transmission in the ferret model. In vitro, both the prodrug and parent drug reduce SARS-CoV-2 titers with minimal cytotoxicity. In the ferret model, the prodrug was administered at 20mg/kg 2x daily starting on the day of the SARS-CoV-2 (Washington strain) challenge. In therapeutic studies where the drug was administered at 12hpi, the 10 mg/kg treatment significantly reduced viral titers. Using the 10 mg/kg therapeutic dosing strategy, the authors examined titers and transmission using the gamma CoV2 strain (BZ/2021) in ferrets. Ferrets were challenged, then 3dpc ferrets were co-housed with naïve ferrets for two days to represent community spread, and contact ferrets were monitored for another 4 days. Treated ferrets had significantly

reduced viral titers against this VoC, and treated ferrets had reduced transmission to contact ferrets, with titers generally at or below LOD. Finally, authors show that treatment did not select for remdesivir-resistant mutations, however, expected ferret-adapted mutations were noted. Overall, this manuscript is well written and presented. The only caveat is authors were unable to show a reduction of disease using this model.

Minor:

- Please note the full name of media used on line 431 - RPMI-1640
- Please state where reagents used for LC-MS/MS were sourced
- Please include a citation for the A549-hACE2 cells used (line 433)
- Better explain EIDD-2801/molnupiravir - this compound is used in one of the figures but only passively described - why it is important to include this compound as a control?
- Please clarify the % viability with the lowest doses of drugs (Figure 1d/e, ext fig 1).
- Please comment regarding the statistical power of $n=2$.

Major comments:

-none

Response to reviews

Reviewer #1 (Remarks to the Author):

In this manuscript the authors profile GS-621763, an oral prodrug of the remdesivir parent previously identified within a discovery program targeting RSV. Here the authors report on GS-621763's PK, its efficacy against SARS-CoV-2 isolates in vitro and its efficacy in prophylactic and therapeutic modes in an in vivo ferret SARS-CoV-2 model of infection. Additionally, the authors report that oral dosing of GS-621763 in SARS-CoV-2-infected ferrets prevents transmission of the virus to untreated contact animals for a variant of concern of the P.1 lineage. They also monitor the mutations in SARS-CoV-2 isolates that emerge during their experiments.

This report is of significant interest, as development of effective oral therapies for COVID-19 would have a positive impact on controlling the current pandemic by improving accessibility to antivirals and potentially limit community spread of the infection.

The manuscript is comprehensive; however, several edits, most minor, should be attended to prior to publication:

1. Additional methods or description of data generation for Extended Data Table 3 is missing. What timepoint were the lungs harvested at (24 h post dose?) and what was measured to determine the "lung total nuc" metric?

The methods supporting Extended Data Table 3 were added to the Materials and Methods section as follows. "Ferret lungs were collected at 24 hours following initiation of drug administration. Whole tissues were quickly isolated and immediately placed into liquid nitrogen and stored at -80°C until processing and LC-MS/MS analysis according to methods previously described (Mackman, et al.). Reported values for lung total nucleosides is the sum of (GS-441524 and mono-, di- and tri-phosphate (GS-443902) metabolites)."

2. No reference or methods for generation of A549-hACE2 cell line is present in the manuscript. These are also not listed in the Reporting Summary. Please add references/methods as appropriate.

A citation for the A549-hACE2 cells has been added.

3. Is there an explanation as to why GS-621763 is less efficacious in vitro against the CA/2020 B.1.1.7 lineage, causing only ~1 log reduction vs ~5 log reduction in titer of other strains?

CA/2020 B.1.1.7 grew less efficiently in vitro and only reached a peak titer of 10^3 pfu/ml. The ~1 log reduction represents virus reduction to the limit of detection (10^2 pfu/ml). Therefore, it is not that GS-621763 is less efficacious, only that this strain reached lower peak titers in vitro.

4. Legends for a couple plots in the manuscript refer to symbols that represent individual biological replicates, but instead show a single (mean or a median) point: Fig. 2a, Extended Data Fig. 1.

Fig. 2a and Extended Data Fig. 1 have been updated to show individual repeats in accordance with journal policy. This error was also present in Fig. 1D and 1E, which have now been updated to show individual repeats.

5. In the introduction, line 59, the authors refer to high oral bioavailability of GS-621763 "in several species". Because "several" technically refers to more than two, and bioavailability seemed to be determined for compound 13 (GS-621763?) in the reference for only rat and cynomolgus monkeys please either list the species or change the wording.

We have clarified in the revised manuscript that high oral bioavailability of GS-621763 was demonstrated in rats and cynomolgus macaques. Technically speaking, “multiple” is defined as more than one, but we have revised the statement, now specifying that “a prodrug strategy to improve oral absorption was employed leading to the identification of GS-621763, which demonstrated high oral bioavailability in two relevant animal species including non-human primates².”

6. Line 33 in abstract: adding that the studies were performed in ferrets would improve readability: e.g., ... to near-undetectable levels in ferrets.

Line 33 in the abstract has been changed as requested. “...to near-undetectable levels in ferrets.”

7. Lines 79 – 81, it is unclear from the results section that EC₅₀s in Vero E6 cells are being discussed. Listing Vero E6 in this section of the text would be helpful.

Change made as requested. “...concentrations (EC₅₀) in SARS-CoV-2 infected Vero E6 cells were highly consistent....”

8. In legend for Figure 3, a comma is missing between vehicle and GS-621763 (line 382).

Change made as requested. A comma was added between vehicle and GS-621763

9. Line 494: “were added” is redundant and an extra closing bracket is missing.

Change made as requested. The phrase “were added” was removed and the extra closing bracket was added.

Reviewer #2 (Remarks to the Author):

Hereby I would like to congratulate the authors of the manuscript “Oral Prodrug of remdesivir parent GS-441524 is efficacious against SARS-CoV-2 and a variant of concern in ferrets” with their nice paper. I am happy that people are looking into the great potential of GS-441524 and was thus content to review the paper! The authors did a great job studying the antiviral effect of their newly developed prodrug (GS-621763) in cell culture and in ferret. The results look promising; however, I lack some information about crucial points.

Major concerns

I'm not entirely convinced that their approach is the most useful. Please comment on the fact that you work with a pro-drug of GS-441524, which still needs to be converted to the active triphosphate, which is a rate-limiting step in vivo. Your compound is basically a new prodrug of GS-441524 (tri-isopropylester prodrug instead of phosphoramidate prodrug that is Remdesivir), with the advantage of a better oral absorption and thus availability. I do agree on the advantage compared to Remdesivir (Oral and high systemic levels of GS-441524), but please highlight why it is better to use this pro-drug, compared to GS-441524 directly, as this compound is also known to be orally available and effective in vivo against SARS-CoV-2.

Seeing this comment, we have better explained our rationale in the revised manuscript. Specifically, the tri-isobutryl ester prodrug GS-621763 of parent nucleoside GS-441524 was

employed in this study because it has superior oral bioavailability across multiple non-clinical species compared to GS-441524. Specifically, in rats and cynomolgus macaques, the oral bioavailability was improved by greater than 5-fold [Mackman et al 2021]. GS-621763 (Cpd 13 in JMC 2021, 64(8), 5001-5017) increased oral F by >5-fold to 57% and 28% in rats and cynomolgus macaques, respectively, compared to only 12% and 3.4%, respectively, to GS-441524 (Cpd 4 in JMC 2021, 64(8), 5001-5017).

Wei et al. (Bioorg. Med Chem Lett 2021, 116364) reached the same conclusion regarding the use of alternate ester prodrugs of GS-441524 in their studies, which demonstrated improved oral bioavailability in mice compared to only 16% oral availability of GS-441524. Also in the SARS-CoV2 mouse model reported by Li et al (J. Med Chem. 2021 doi.org/10.1021/acs.jmedchem.0c01929). GS-441524 was administered by intraperitoneal injection because oral bioavailability was measured at a poor 5%.

So far, high oral availability of GS-441524 was only reported in dogs (89%). However, dogs are considered an outlier species in this context, since this unusual result is thought to be due to the known 'leaky' paracellular junctions in dogs, which are not representative of human oral properties. Human dosing of GS-441524 was reported in a single individual (clinicaltrials.gov NCT04859244) and the target plasma concentrations of GS-441524 required a high total daily dose of 2,250 mg, again consistent with low oral bioavailability. *In toto*, published data from multiple independent groups provide sound evidence that GS-441524 has low oral bioavailability, and that the ester prodrug approach results in improved oral properties.

We have included the above facts and references in the revised manuscript to provide the reader with the relevant context that forms the basis for our study.

Please comment on other possible chemical options, e.g. a prodrug which easily converts to GS-443902 or an oral Remdesivir analogue (although known to be very difficult)?

The phosphoramidate prodrugs are not well-suited for oral delivery due to limited absorption and high first pass extraction that reduces systemic exposure of prodrug. However, oral phospholipid prodrugs of remdesivir that deliver the same monophosphate metabolite have recently been reported [Schooley AAC 2021], suggesting that other phosphate prodrug avenues may exist for oral delivery of the remdesivir anabolite GS-443902. However, the objective of our study was to assess antiviral efficacy of GS-621763 in a relevant animal model of SARS-CoV-2 infection. We prefer to keep this focus.

Similarly, please compare the effect of your pro-drug in vivo with the effect of oral administration of GS-441524 directly. This will highlight to the advantage of your prodrug (higher exposure of GS-441524) compared to GS-441524 itself.

As referenced above, GS-621763 has demonstrated superior oral bioavailability compared to GS-441524. We consider it unethical to commit large research animals to address a question that is superseded by the available published literature and has little, if any, practical relevance.

PK should include measurement of GS-443902 (in PBMC) for all compounds/routes.

It is unclear to us what can be learned from triphosphate measurements in PBMCs. As also requested by the reviewer, we have added efficacy testing of GS-441524 and GS-621763 in air-liquid interface cultures of primary human airway cells (see below), since these organoids represent a disease relevant tissue model. Potent antiviral activity in the model confirms efficient conversion to GS-443902 in primary human tissues.

I appreciate the study with delayed start of treatment, however, I believe 12 hours is not enough. Please comment on further delay (24 – 48h) of treatment start, as this would be more relevant in a clinical setting.

SARS-CoV-2 disease progression is faster in ferrets than humans. Twelve hours post infection was selected based on shed virus becoming first detectable in nasal lavages. Since the onset of the COVID-19 pandemic, groundbreaking changes in the diagnostic testing infrastructure has allowed for early detection of infection based on rapid PCR tests after community contact tracing. These changes to diagnostic capabilities have created a scenario where treatment of humans can be initiated at a comparable disease state, provided the therapeutic is orally available.

We have modified text to add clarity on the rationale for choosing 12 hours post infection as an appropriate start of treatment. Line 119: "...12 hours after infection, when shed virus is first detectable in nasal lavages..."

Besides all the in vivo work, it maybe not highly relevant, but it would be nice to see antiviral data in multiple cell types instead of only VeroE6 cells. I encourage the group to verify antiviral effect especially in primary HAE cells to have more human data.

We agree with the reviewer and have provided antiviral efficacy data for GS-441524 and GS-621763 on well-differentiated primary human airway epithelium tissue models grown at air-liquid interface against a SARS-CoV-2 variant of concern (BZ/2021). This important new data has been incorporated in Figure 1.

Minors concerns

Improve the picture (Fig. 1) to describe Remdesivir metabolization in human (monophosphate metabolite and anabolization to GS-443902).

We have included additional references (line 53) on the metabolism of remdesivir to the active triphosphate GS-443902, since metabolism of remdesivir in humans has been extensively studied and published. The objective of Figure 1 was to provide an overview visualizing that GS-621763 is converted to GS-441524 after oral administration, followed by conversion to bioactive GS-443902 in the tissues.

Remove all P values from the picture to make it more ordered/easier to view.

Inclusion of P values in figures is a requirement of Nature Communications. Following journal instructions, P values were stated in all figures.

In the transmission experiment: after 52 hours, there is already a great reduction of infectious virus in the ferret (as shown in the previous experiments). So, maybe it would be better to co-house the animals in an earlier stage (when there is still a lot of virus present).

We want to emphasize that all contact ferrets were left untreated, which we consider the clinically relevant situation (treatment of the infected source, no prophylactic treatment of the uninfected sentinels). If untreated contact ferrets are cohoused with treated source ferrets that are still shedding "a lot of virus," naturally rapid transmission will occur. The objective of our experiment was to address whether treatment renders an infected source faster non-contagious than vehicle, which was indeed the case.

Although I still believe in the effect of the compound, I think the transmission experiment per se is not as relevant as stated. (patients that are being treated are aware that they are infected and thus quarantine) More relevant would be later start of treatment (more than the 12hours in the paper here), as mentioned before.

The transmission study was designed to simulate the effects of drug treatment on individuals recently exposed and infected with SARS-CoV-2. It addresses the effect of treatment on duration of virus shedding and spread compared to untreated individuals. This question is highly relevant, since current CDC guidelines suggest that patients quarantine until 10 days after a positive PCR test (<https://www.cdc.gov/coronavirus/2019-ncov/your-health/quarantine-isolation.html>) which causes considerable distress. Pharmacological shortening of quarantine periods would improve quality of life of the infected and reduce opportunity for transmission in the communities, potentially making an important contribution to silencing the pandemic.

Misspelling of the pro-drug in the legend of Fig. 1 ('GS-627163 instead of GS-621763)?

We have corrected the typo.

The authors claim there is little escape variants after *in vivo* treatment, however, a study is only 4 days. I feel this should be backed by long term cell culture experiments or longer *in vivo* studies.

It was our objective to address whether viral escape can occur rapidly during the primary infection stage *in vivo*. In duration, the 4-day infection cycle of SARS-CoV-2 is similar to that with many influenza virus models. In clinical trials, resistance to the influenza virus inhibitor baloxavir marboxil, for instance, emerged rapidly in humans. We therefore considered it important to test for characteristic changes in compound-experienced viruses recovered from infected animals. Later timepoints could not be assessed because virus becomes rapidly undetectable in treated ferrets, and ethical concerns and cost are prohibitive to consecutive viral passaging in ferrets.

Long term cell culture viral adaptations to remdesivir have been published or are available as preprints. Since GS-443902 is the bioactive anabolite of both remdesivir and GS-621763, it is unclear what new is anticipated to be learned from repeating published work.

Please specify how temperature is measured. I'm not sure if this is a relevant factor, as in Fig. 3,D it shows no temperature elevation in vehicle treated animals.

The method of ferret temperature reading has been added to the methods. Ferret temperatures were measured rectally. Ferrets display little to no clinical signs of SARS-CoV-2 infection, similar to healthy young adult populations. Although vehicle treated ferrets in figure 1D show a significant difference in temperature compared to treated ferrets, only one animal reaches an internal temperature that can be considered a low grade fever ($>39.5^{\circ}\text{C}$). However, inclusion of this data may be important for future cross-study comparisons, since infection with newly emerging VOC might lead to the appearance of additional clinical signs in the ferret model.

If this prodrug is suitable for clinical studies, please comment on production expectation, stability, storage and if you expect any toxicity issues.

Questions concerning production, stability of the drug product, shelf live etc. become relevant at clinical trial stage and/or product launch, but are premature in the context of our study.

Fig. 2,H shows 10^7 RNA copies in nasal turbinate in vehicle treated, while Fig. 3,G shows 10^9 copies. Is this expected variation?

Ferrets are outbred animals and inherently a greater amount of variation compared to inbred (i.e. mouse) models must therefore be anticipated. The clinical experience with SARS-CoV-2 demonstrates considerable variation in (outbred) humans also, underscoring relevance of the animal model.

Reviewer #3 (Remarks to the Author):

This study examined the efficacy of the oral prodrug (GS-621763) of remdesivir parent (GS-441524) to reduce COVID-19 viral titers and transmission in the ferret model. In vitro, both the prodrug and parent drug reduce SARS-CoV-2 titers with minimal cytotoxicity. In the ferret model, the prodrug was administered at 20mg/kg 2x daily starting on the day of the SARS-CoV-2 (Washington strain) challenge. In therapeutic studies where the drug was administered at 12hpi, the 10 mg/kg treatment significantly reduced viral titers. Using the 10 mg/kg therapeutic dosing strategy, the authors examined titers and transmission using the gamma CoV2 strain (BZ/2021) in ferrets. Ferrets were challenged, then 3dpc ferrets were co-housed with naïve ferrets for two days to represent community spread, and contact ferrets were monitored for another 4 days. Treated ferrets had significantly reduced viral titers against this VoC, and treated ferrets had reduced transmission to contact ferrets, with titers generally at or below LOD. Finally, authors show that treatment did not select for remdesivir-resistant mutations, however, expected ferret-adapted mutations were noted. Overall, this manuscript is well written and presented. The only caveat is authors were unable to show a reduction of disease using this model.

Minor:

- Please note the full name of media used on line 431 - RPMI-1640

Change made as requested.

- Please state where reagents used for LC-MS/MS were sourced

Sources for LC-MS/MS reagents have been included in the Methods section.

- Please include a citation for the A549-hACE2 cells used (line 433)

A reference for the A549-hACE2 cells has been added (reference 20 of the revised manuscript).

- Better explain EIDD-2801/molnupiravir - this compound is used in one of the figures but only passively described - why it is important to include this compound as a control?

We have better explained our rationale for including EIDD-2801 as a reference in the revised manuscript. "EIDD-2801 was chosen as a benchmark since at the time of this study, it is the only orally available nucleoside analog that has shown efficacy against SARS-CoV-2 in the ferret model."

- Please clarify the % viability with the lowest doses of drugs (Figure 1d/e, ext fig 1).

We suspect an edge effect present in the 96-well dose response plates. The variability seen at the lowest compound concentration is likely a consequence of increased evaporation of tissue culture media on outside regions of the culture plate during plate incubation for 72 hours.

- Please comment regarding the statistical power of $n=2$.

For this revision, we have performed and added additional experiments to the Extended Data Table 1, now increasing the number of independent replicates to four (potency tests against WA1/2020-nano luciferase) and six (cytotoxicity test), respectively. Statistical analyses and the source data file were updated accordingly.

Major comments:

-none

Reviewers' Comments:

Reviewer #2:

Remarks to the Author:

Dear Authors,

thank you for thoughtfully answering all my questions and remarks! I have no further objections against accepting the paper in Nature communications. Thank you for your nice work and good luck with further research.

Response to reviews

REVIEWERS' COMMENTS

Reviewer #2 (Remarks to the Author):

Dear Authors,

thank you for thoughtfully answering all my questions and remarks! I have no further objections against accepting the paper in Nature communications. Thank you for your nice work and good luck with further research.

There were no reviewer comments to address.